# Computational Modeling of the Chlamydial Developmental Cycle Reveals a Potential Role for Asymmetric Division

Travis J. Chiarelli,[a] Nicole A. Grieshaber,[a] Cody Appa,[a] (iD) Scott S. Grieshaber[a]

[a]Department of Biological Sciences, University of Idaho, Moscow, Idaho, USA

**ABSTRACT** *Chlamydia trachomatis* is an obligate intracellular bacterium that progresses through an essential multicell form developmental cycle. Infection of the host is initiated by the elementary body (EB). Once in the host, the EB cell differentiates into the noninfectious, but replication-competent, reticulate body, or RB. After multiple rounds of replication, RBs undergo secondary differentiation, eventually producing newly infectious EBs. Here, we generated paired cell-type promoter reporter constructs and determined the kinetics of the activities of the *euo*, *hctA*, and *hctB* promoters. The paired constructs revealed that the developmental cycle produces at least three phenotypically distinct cell types, the RB (*euo*prom+), intermediate body (IB; *hctA*prom+), and EB (*hctB*prom+). The kinetic data from the three dual-promoter constructs were used to generate two computational agent-based models to reproduce the chlamydial developmental cycle. Both models simulated EB germination, RB amplification, IB formation, and EB production but differed in the mechanism that generated the IB. The direct conversion and the asymmetric production models predicted different behaviors for the RB population, which were experimentally testable. In agreement with the asymmetric production model, RBs acted as stem cells after the initial amplification stage, producing one IB and self-renewing after every division. We also demonstrated that IBs are a transient cell population, maturing directly into EBs after formation without the need for cell division. The culmination of these results suggests that the developmental cycle can be described by a four-stage model, EB germination, RB amplification/maturation, IB production, and EB formation.

**IMPORTANCE** *Chlamydia trachomatis* is an obligate intracellular bacterial pathogen responsible for both ocular and sexually transmitted infections. All *Chlamydiae* are reliant on a complex developmental cycle, consisting of both infectious and noninfectious cell forms. The EB cell form initiates infection, whereas the RB cell replicates. The infectious cycle requires both cell types, as RB replication increases the cell population while EB formation disseminates the infection to new hosts. The mechanisms of RB-to-EB development are largely unknown. Here, we developed unique dual promoter reporters and used live-cell imaging and confocal microscopy to visualize the cycle at the single-cell and kinetic levels. These data were used to develop and test two agent-based models, simulating either direct conversion of RBs to EBs or production of EBs via asymmetric RB division. Our results suggest that RBs mature into a stem cell-like population producing intermediate cell forms through asymmetric division, followed by maturation of the intermediate cell type into the infectious EB. Ultimately, a more complete mechanistic understanding of the developmental cycle will lead to novel therapeutics targeting cell type development to eliminate chlamydial dissemination.

**KEYWORDS** chlamydia, differentiation, computational modeling, cell cycle, cell differentiation, cellular development, computer modeling, gene expression

Address correspondence to Scott S. Grieshaber, scottg@uidaho.edu.

The authors declare no conflict of interest.

**C**hlamydiae are obligate intracellular bacterial parasites that cause an array of diseases in both humans and animals. *Chlamydia trachomatis*, a human-adapted pathogen, is the leading global cause of bacterial sexually acquired infections and preventable blindness. In

2019, the CDC reported 1.8 million *C. trachomatis* infections within the United States alone, with the most recent reports indicating that rates increased by 10.0% in women and 32.1% in men from 2015 to 2019 (1, 2). This increase in infection rates has been reported across all racial/ethnic groups and affects all age groups (2).

Chlamydial growth and development have classically been characterized as a biphasic cycle, consisting of two primary cell forms, the elementary and reticulate body (3). These cell forms maintain a division of labor throughout the infectious cycle and are essential for chlamydial proliferation. The elementary body (EB) is the infectious cell form and initiates host cell invasion by pathogen-mediated endocytosis (4). The EB cell form is nonreplicative, and the chromosome is tightly compacted by nucleoid-associated proteins (5, 6). Upon entry into the host, the EB undergoes large transcriptional and phenotypic changes, maturing into the reticulate body (RB) in a process that takes up to 12 h for serovar L2 (7, 8). The RB is replication competent but noninfectious and must redifferentiate back into the EB to disseminate the infection to new host cells (3, 9).

Electron micrographs have also shown the presence of a transitory cell form, termed the intermediate body (IB). IBs are present beginning between 20 and 24 h postinfection (hpi) for serovar L2 and are characterized by a semicondensed nucleoid similar to the EB, but they are significantly larger, which gives them a target-like appearance (7, 10). Due to the presence of the IB and its appearance as a transitory form, it is currently hypothesized that a subset of RBs undergoes large morphological changes to convert directly into EBs.

We previously reported the development of a live-cell reporter system to follow the chlamydial cycle in real time at the single-inclusion level. A suite of different gene promoters was designed to drive the expression of fluorescent proteins in order to follow RB growth and EB development. These kinetic data suggested that the promoters fell into three temporal categories exemplified by the activity of the *euo*, *hctA*, and *hctB* promoters (10).

In this study, paired promoter reporter constructs were developed to determine the temporal and spatial relationships between the activities of the *euo*, *hctA*, and *hctB* promoters at the kinetic and single-cell levels. Based on the expression kinetics of these reporters and their kinetic relationships to each other, computational agent-based models were created to best represent the developmental cycle. Two models were developed to explain the data, an RB-to-IB direct conversion model and an asymmetric division/production model. The outputs of simulations from these models were compared to experimental data to determine which mechanism was best supported. Our model and data suggest a novel RB amplification/maturation step where the RB initially divides symmetrically to produce two RBs and increase RB numbers, followed by maturation of the RB to an asymmetrically dividing cell that produces one IB while regenerating the RB. Our data also support the direct maturation of the IB cell type into the EB without the need for cell division.

## RESULTS

**Development of dual fluorescent cell reporter constructs to determine cell-type gene expression during the developmental cycle.** We previously reported the use of promoter reporter constructs that were expressed at different stages of the chlamydial developmental cycle (10). Fluorescence from the *euo*prom-Clover construct was first detected at ~14 h postinfection (hpi) and demonstrated a short exponential phase, followed by an expression plateau at ~24 hpi (10). Imaging of the *hctA*prom-Clover construct revealed that the Clover signal could first be detected at ~18 hpi and continued to increase until cell lysis (10). Additionally, we noted that *hctB*prom-Clover expression was initiated at ~22 hpi, ~4 h later than *hctA*prom, and also increased linearly until cell lysis (10).

Here, we extended these studies by combining these promoters in pairs on the same plasmid. In addition, the LVA degradation tag was used to target fluorescence proteins for degradation to increase temporal resolution. To visualize expression kinetics and expression relationships between these cell-type promoter reporters at the single-inclusion and single-cell level, we constructed three dual fluorescent developmental gene expression reporter strains, *hctA*prom-*euo*prom, *hctB*prom-*hctA*prom, and *hctB*prom-*euo*prom. For the *hctA*prom-*euo*prom (AMELVA) construct, the *euo* promoter was used to drive the expression of the green fluorescent protein variant, mNeonGreen (mNG) (11) fused in-frame to

**TABLE 1** List of reporter strains and their relevant promoter constructs

| Strain | Promoter reporter 1 | Promoter reporter 2 | Overexpressed protein | Cell type(s) |
|---|---|---|---|---|
| L2-BMEC | *hctB*prom-mKate2 | *euo*prom-Clover | | All cells (green), EBs (red) |
| L2-BMELVA | *hctB*prom-mKate2 | *euo*prom-neongreen(LVA)[a] | | RBs (green), EBs (red) |
| L2-AMELVA | *hctA*prom-mKate2 | *euo*prom-neongreen(LVA)[a] | | RBs (green), IBs (red) |
| L2-BMALVA | *hctB*prom-mKate2 | *hctA*prom-neongreen(LVA)[a] | | IBs (green), EBs (red) |
| L2-BMAMEO | *hctB*prom-mKate2 | *hctA*prom-mEos | | IBs and EBs (green), EBs (red) |
| L2-E-*ftsI*-BMAMEO | *hctB*prom-mKate2 | *hctA*prom-mEos | FtsI | IBs and EBs (green), EBs (red) |
| L2-E-*ftsI*-BMELVA | *hctB*prom-mKate2 | *euo*prom-neongreen(LVA)[a] | FtsI | RBs (green), EBs (red) |

[a]Fluorescent half-life, 30 min.

the LVA protein degradation tag, which reduced the fluorescent half-life to ~30 min (12), and the *hctA* promoter was used to drive the expression of the red fluorescent protein mKate2 (13) (see Fig. S1A in the supplemental material). The *hctB*prom-*euo*prom (BMELVA) dual-reporter construct was created by replacing *hctA*prom in AMELVA with *hctB*prom to drive mKate2 expression (Fig. S1B). Last, to create the *hctB*prom-*hctA*prom (BMALVA) dual reporter, the *hctA* promoter was used to drive mNG(LVA), and the *hctB* promoter was used to drive mKate2 expression (Fig. S1C). Each construct was transformed into *Chlamydia trachomatis* L2, creating the L2-AMELVA, L2-BMELVA, and L2-BMALVA reporter strains (Table 1) and compared to our previously reported dual-promoter strain L2-*hctB*prom_mKate2-*euo*prom_Clover (L2-BMEC) (10). To determine the relationships between the promoter reporters, we investigated the expression of each paired reporter at the single chlamydial cell level by confocal microscopy (Fig. 1A), and the single inclusion kinetic level by live-cell imaging (Fig. 1B). For confocal imaging, host cells were infected with each strain for 24 h followed by fixation. All L2-BMEC chlamydial cells showed some level of green fluorescence, suggesting that all cells expressed from the *euo*prom-Clover construct at some point during the developmental cycle (Fig. 1A, L2-BMEC). However, confocal microscopy of the new strains using the short-half-life mNG(LVA) fluorescent reporter revealed that the three promoters were active exclusively from one another (Fig. 1A). The *euo*prom-mNG(LVA) signal from the L2-AMELVA strain was present in large RB-sized cells, while *hctA*prom-mKate2 was active in a subset of large and small cells but in a population distinct from *euo*prom⁺ cells (Fig. 1A, L2-AMELVA). The *euo*prom-mNG(LVA) signal from L2-BMELVA was also present in large cells only, while the *hctB*prom-mKate2 signal was present in small cells and in an entirely distinct population (Fig. 1A, BMELVA). In the L2-BMALVA-infected cells, *hctA*prom-mNG(LVA) expression was visible in large and small cells, while the *hctB*prom-mKate2 signal was detected in small cells. Again, the two promoter reporters were active in distinct, nonoverlapping populations (Fig. 1A, L2-BMALVA). Occasionally, we saw paired *hctA*prom-expressing cells, but we do not know the origin or fate of these cells, as these are static images of a dynamic environment.

We quantified the expression of the paired fluorescent reporters at the single-cell level in four individual inclusions for each strain. The TrackMate plugin in Fiji (14) was used to identify green fluorescent protein (GFP)- and red fluorescent protein (RFP)-expressing cells and record the fluorescent signal for each reporter from entire confocal z stacks encompassing each inclusion. The paired fluorescent signals for each cell were plotted, the RFP on the *y* axis and the GFP signal on the *x* axis (Fig. 1A). For the L2-BMEC strain expressing the stable GFP protein, Clover, driven by the *euo* promoter, both green and red cells had Clover signal, demonstrating that both cell types expressed Clover from the *euo* promoter at some point during the developmental cycle of that cell. In contrast, the red- and green-expressing *Chlamydia* cells of the L2-BMELVA strain, which expresses a short-half-life fluorescent protein from the *euo* promoter, are distinct populations with minimal overlap. The same was seen for the L2-AMELVA and L2-BMALVA infections; the red-expressing cells had minimal green signal (AMELVA), and the green-expressing cells had minimal red signal (BMALVA). Together, these data demonstrate that the three promoters are active in distinct subpopulations of chlamydial cells.

In addition to the confocal images, the kinetics of each promoter was determined

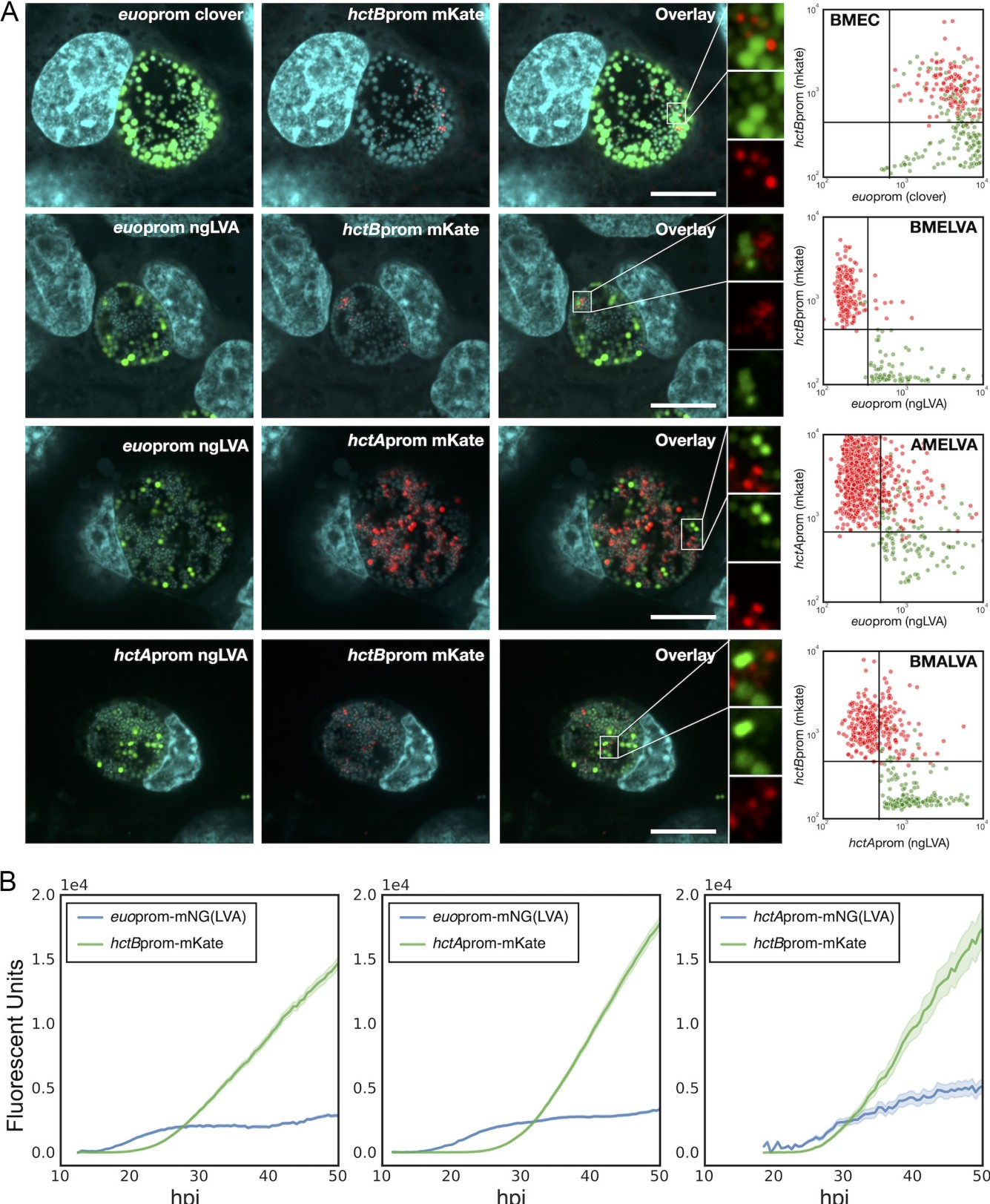

**FIG 1** Dual cell form-specific promoter reporter chlamydial strains reveal three distinct chlamydial cell populations. (A) Representative single z-plane confocal micrographs of Cos-7 cells infected with Ctr-L2-*hctB*prom-mKate2_*euo*prom-Clover (L2-BMEC), Ctr-L2-*hctB*prom-mKate2_*euo*prom-mNG(LVA) (L2-BMELVA), Ctr-L2-*hctA*prom-mKate2_*euo*prom-mNG(LVA) (L2-AMELVA), and Ctr-L2-*hctB*prom-mKate2_*hctA*prom-mNG(LVA) (L2-BMALVA) reporter strains. The infections were fixed at 24 hpi and imaged. For L2-BMEC, Clover (green), mKate2 (red), and DAPI (cyan) images are shown. For L2-BMELVA, L2-AMELVA, and L2-AMALVA, mNG(LVA)

at the single-inclusion level using live-cell microscopy. Host cells were infected with each strain and imaged for both GFP and RFP from 10 hpi until 50 hpi at 30-min intervals. *Euo*prom-mNG(LVA) signal from L2-AMELVA was first detected at ~14 hpi and increased exponentially until ~26 hpi, after which time the signal reached a plateau that was maintained for the duration of the infection (Fig. 1B, L2-AMELVA). The *hctA*prom(mKate2) signal was first detectable at ~18 hpi, with an exponential increase in expression until 28 hpi, followed by a linear increase until the end of the experiment at 50 hpi (Fig. 1B, L2-AMELVA). Like L2-AMELVA, the *euo*prom-mNG(LVA) signal for L2-BMELVA followed the same kinetics, with an early exponential increase followed by a signal plateau (Fig. 1B, L2-BMELVA). The *hctB*prom signal in these inclusions became detectable at ~24 hpi and increased exponentially until ~34 hpi. After this brief exponential phase, *hctB*prom(mKate2) signal increased at a linear rate until the end of the experiment (Fig. 1B, L2-BMELVA). Live-cell kinetics of L2-BMALVA showed that *hctA*prom-mNG(LVA) activity initiated around 18 hpi; however, expression approached steady-state kinetics and did not accumulate (Fig. 1B, L2-BMALVA). *HctB*prom(mKate2) demonstrated the same kinetics as the L2-BMELVA strain (Fig. 1B). The short half-life of the LVA-tagged mNG allowed for spatial and kinetic resolution of three cell types. These results demonstrate that chlamydial cells follow a temporal gene expression program resulting in three distinct cell forms. The RB cell activity expresses from the *euo*prom, while the EB cell expresses from the *hctB*prom. The observation that *hctA*prom was temporarily active after *euo*prom and before *hctB*prom and the observation that *hctA*prom is active in a distinct cell population from *euo*prom-positive (*euo*prom$^+$) cells and *hctB*prom$^+$ cells suggest that *hctA*prom is active in the IB cell population. Overall, these data suggest that the developmental cycle can be represented by three phenotypically distinct cell types, RB cells (*euo*prom$^+$), IB cells (*hctA*prom$^+$), and EB cells (*hctB*prom$^+$).

Live-cell imaging also revealed the interrelated kinetics of the activity of these promoters. The kinetic data from each individual inclusion of both L2-AMELVA (Fig. 2A) and L2-BMELVA (Fig. 2B) revealed that there was significant heterogeneity in the maximal expression level of the *euo*prom plateau. Interestingly, the variation in *euo*prom expression from the paired expression constructs correlated with the rates of *hctA*prom (L2-AMELVA) and *hctB*prom (L2-BMELVA) signal accumulation, i.e., inclusions exhibiting high *euo*prom plateaus had steeper slopes for *hctA*prom and *hctB*prom signal accumulation, while lower *euo*prom signal correlated with a lower rate of *hctA*prom and *hctB*prom signal accumulation (Fig. 2A and B). When the *hctA*prom and *hctB*prom signals were normalized to the *euo*prom signal plateau for each inclusion, the variation in the slopes of *hctA*prom(mKate2) and *hctB*prom(mKate2) accumulation was dramatically reduced (Fig. 2A and B). This variability in the maximum expression levels of *euo*prom suggested that each inclusion contained differing numbers of RB (*euo*prom$^+$) cells during the plateau phase, which, in turn, led to the various IB (*hctA*prom) and EB (*hctB*prom) accumulation rates.

To test whether the differing plateau signal of *euo*prom expression was due to differing RB numbers, cells were infected with L2-BMELVA at a multiplicity of infection (MOI) of ~0.1 and fixed and stained with DAPI (4′,6-diamidino-2-phenylindole) every 2 h from 14 to 48 hpi. Infected cells were imaged by three-dimensional (3D) confocal microscopy, and cell type quantification was carried out using an automated cell counting workflow using the open-source software Fiji and the TrackMate plugin to count individual *Chlamydia* based on fluorescent reporter intensity (14). These experiments revealed that RB (*euo*prom$^+$) cells increased in number from 14 hpi to 26 hpi, reaching an average of ~30/inclusion (Fig. 2C). After this time point, the average number of RBs was unchanged. However, it was clear that

**FIG 1** Legend (Continued)

(green), mKate2 (red), and DAPI (cyan) images are shown. Magnified fields of view (FOVs) for L2-BMELVA, L2-AMELVA, and L2-BMALVA demonstrate cell form-specific expression in individual cells, while the L2-BMEC-expressing chlamydial cells show green-only cells and red and weakly green cells. Scale bar, 10 $\mu$m. Quantification of both the GFP and RFP channels of GFP [Clover or mNG(LVA)] and mKate2-positive cells was performed using TrackMate from 4 individual inclusions per strain. The red dots each represent a single RFP-expressing cell, while the green dots represent GFP-expressing cells. The measured fluorescent values from each channel for both cell types were plotted (*y* axis, RFP, and *x* axis, GFP fluorescent signal). Chlamydial cells from L2-BMEMC infections demonstrate overlapping GFP and RFP expression in RFP-positive cells, while RFP- and GFP-expressing cells are distinct populations for L2-BMELVA, L2-AMELVA, and L2-BMALVA. (B) Live-cell expression kinetics of L2-BMELVA, L2-AMELVA, and L2-BMALVA reporter strains from *n* > 20 individual inclusions. Infections were imaged from 10 to 50 hpi via automated live-cell fluorescence microscopy. Average intensities are shown; cloud represents SEM.

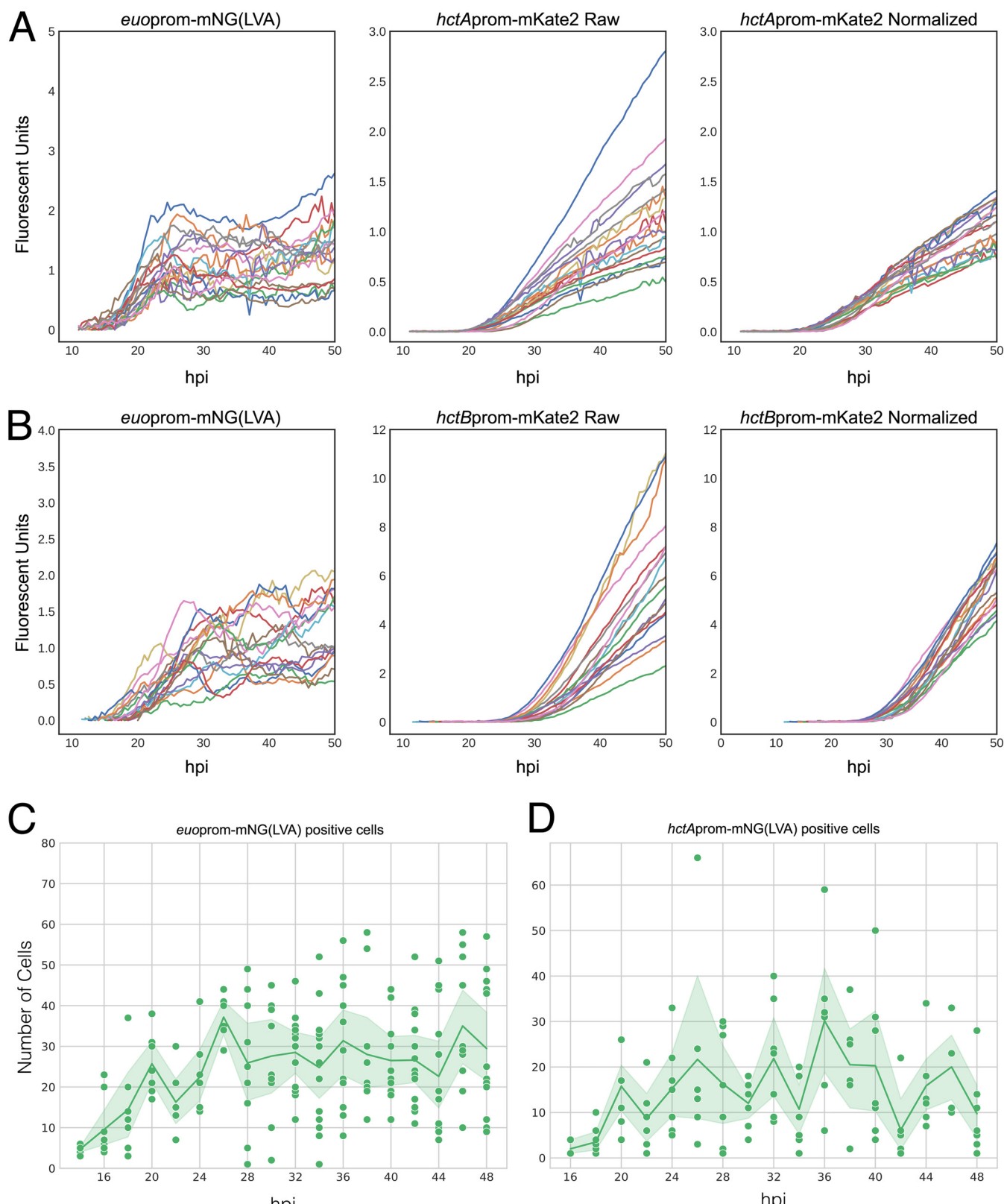

**FIG 2** IB and EB production is dependent on the number of RBs. Cos-7 cells were infected with purified *Ctr*-L2-prom EBs. (A) Live-cell expression kinetics from single inclusions from L2-AMELVA-infected cells. Graphs show *euo*prom-mNG(LVA) expression, *hctA*prom-mKate2 expression, and *hctA*prom-mKate2 expression normalized to the paired average expression levels of *euo*prom-mNG(LVA) between 30 and 38 hpi. (B) Live-cell expression kinetics from single inclusions of L2-BMELVA-infected cells. Graphs show *euo*prom-mNG(LVA) expression, *hctB*prom-mKate2 expression, and *hctB*prom-mKate2 expression normalized to the paired

the maximum number of RBs in each inclusion varied significantly, with some inclusions having as few as one *euo*prom$^+$ cell, while others had as many as 59 RBs/inclusion during this plateau phase (26 to 48 hpi) (Fig. 2C). A similar time course was carried out for cells infected with L2-BMALVA, with similar results. IBs (*hctA*prom$^+$) cells increased in number from 17 hpi until reaching a maximum at 32 hpi, after which the average of IBs/inclusion remained steady (Fig. 2D). Again, similar to the *euo*prom$^+$ cells, the number of *hctA*prom$^+$ cells was significantly different on a per-inclusion basis, with some inclusions having as few as a single *hctA*prom$^+$ cell, while others had as many as 65 during the plateau phase (Fig. 2D). These data suggest that the chlamydial developmental cycle produces significant heterogeneity between inclusions, but despite this heterogeneity, each inclusion produces similar kinetic relationships between cell types.

**Modeling the chlamydial developmental cycle.** Dissecting the mechanisms that control the developmental cycle in *Chlamydia* has, in part, been inhibited by the reliance on population-based studies. Using the individual inclusion kinetic data and individual cell expression data generated from L2-AMELVA, L2-BMELVA, and L2-BMALVA, we divided the cycle into discrete steps, EB germination, RB amplification, IB formation, and EB maturation. To explore potential mechanisms involved in these steps, two agent-based models (ABMs) were developed to describe the developmental cycle using the Python-based bacterial growth simulation platform, Cellmodeller (15). Euo, HctA, and HctB protein expression was simulated in each cell type over time for both models (Text S1). Additionally, germination was set at 10 hpi, as the first replication event has been previously reported at this time point (10, 16), and it agrees well with the initiation of *euo* promoter expression (Fig. 1 and 2) (10). The two models differed in the mechanism controlling RB amplification and IB formation. The asymmetric production model used an RB maturation mechanism where early RBs (designated the RB$_R$) replicate to produce two identical RB$_R$ daughter cells, resulting in RB$_R$ number amplification. This step is followed by an RB$_R$ maturation phase where the RB$_R$ matures over time into a cell form that undergoes asymmetric cell division (designated the RB$_E$), producing one RB$_E$ daughter cell and one IB daughter cell. The direct conversion model used a stochastic direct conversion mechanism where early RBs replicate, resulting in RB number amplification followed by an increase in the chance that an RB transitions into the IB state. This stochastic chance of conversion increases over time before reaching a maintenance state where RB amplification and conversion rates are matched (Text S1). Both models reproduced the developmental kinetics that were observed using L2-AMELVA, L2-BMELVA, and L2-BMALVA (Fig. 1 and 2; Fig. 3A). However, the stochastic model needed to be constrained to match the experimental data. When conversion became greater than RB replication, RB numbers dropped to extinction. Conversely, when the RB replication rate remained higher than the conversion rate, RBs quickly outnumbered EBs (Fig. 3B).

Although both models could produce similar kinetics at the population level, simulations of individual inclusions demonstrated large kinetic differences. For the asymmetric production model, simulated inclusions with high RB numbers had corresponding high EB production rates, while inclusions with low RB numbers had corresponding low EB production rates. The EB production rate for each inclusion, regardless of RB numbers, was linear, while the RB population numbers remained unchanged over time (Fig. 3C, asymmetric production). The kinetics for the direct conversion model had similar trends; however, there were obvious runs of over- and underamplification/conversion for both RB and EB numbers in the individual inclusion simulations, demonstrating that the stochastic mechanism can reliably reproduce the observed data only on the population level. These simulations suggest that the asymmetric production model is a better match for the observed developmental kinetics.

**FIG 2** Legend (Continued)
average expression levels of *euo*prom-mNG(LVA) between 30 and 38 hpi. $n > 20$ inclusions per treatment. (C and D) Quantification of *euo*prom$^+$ or *hctA*prom$^+$ cell counts within individual inclusions from fixed samples. Infections were fixed every 2 h from 14 to 48 hpi and stained with DAPI. Inclusions were imaged by confocal microscopy for DAPI, GFP, and RFP. Individual dots represent the number of promoter reporter-positive chlamydial cells within individual inclusions. Solid line represents the mean number of promoter reporter-positive cells per time point. Sample size ranged between 3 and 14 inclusions, dependent on the time point. Cloud represents 95% confidence interval (CI).

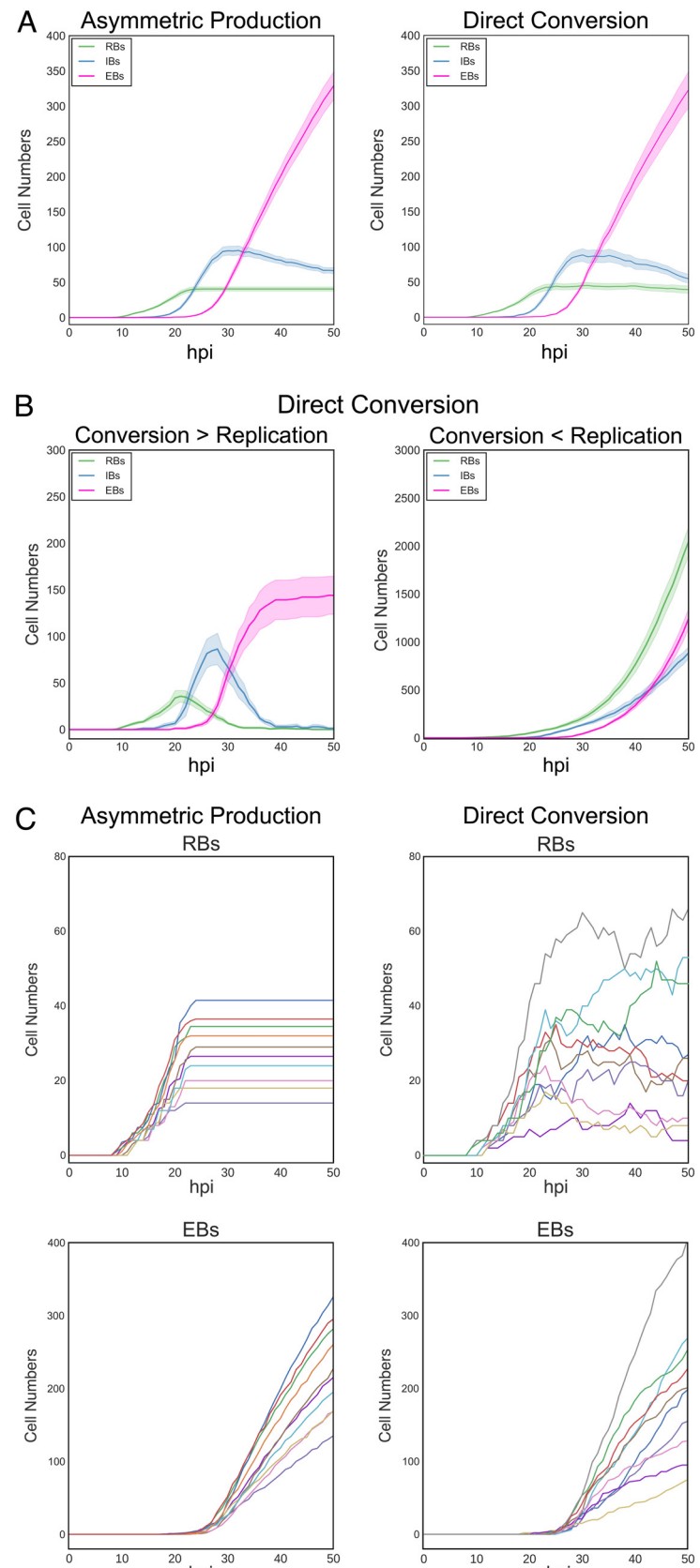

**FIG 3** Simulations of inclusion kinetics from the asymmetric production and direct conversion models. (A) Simulated developmental kinetics of the asymmetric production and direct conversion models. RBs, green;

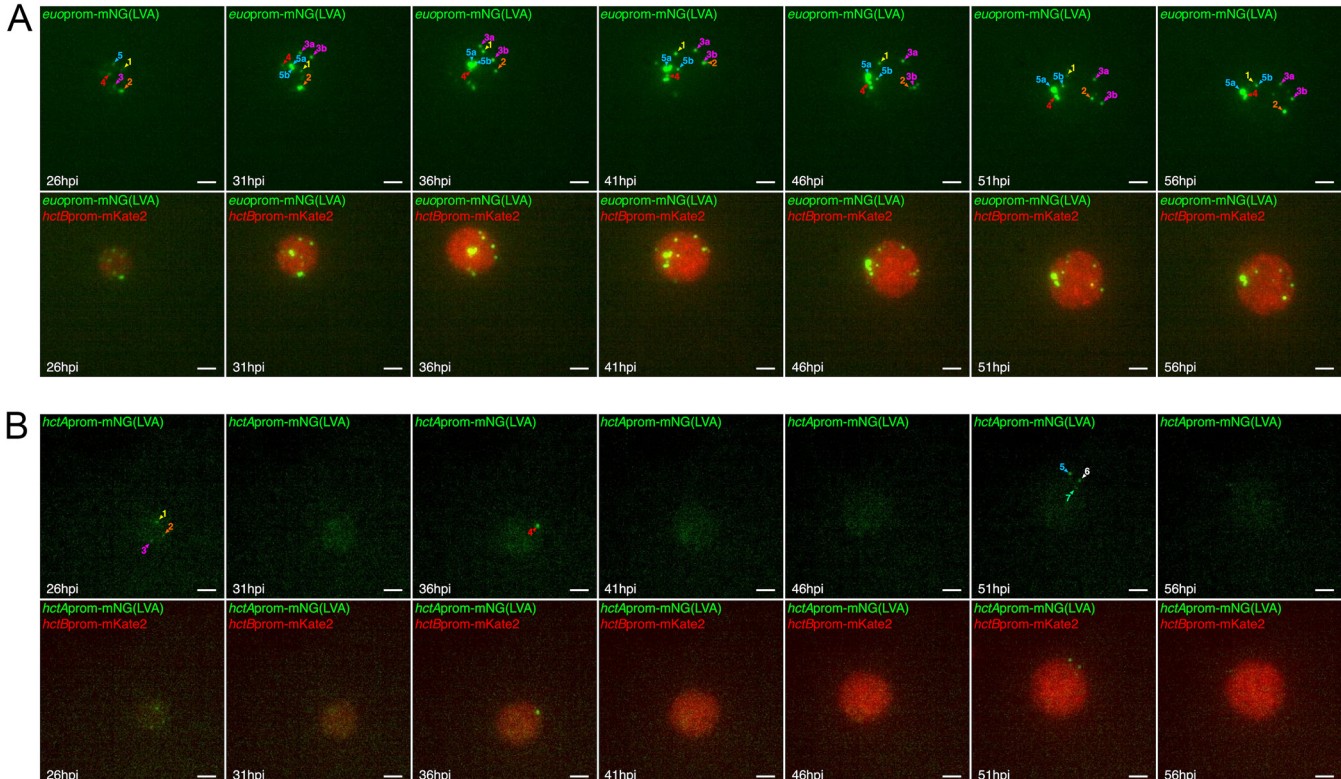

**FIG 4** RBs persist as a stem cell population while IBs are at steady state. Cos-7 cells were infected with either L2-BMELVA (RB/EB) (A) or L2-BMALVA (IB/EB) (B). Individual inclusions were imaged using a 40× objective every 15 min from 24 to 60 hpi. Numbered arrowheads indicate individual *euo*prom-mNG(LVA)$^+$ (RB) or *hctA*prom-mNG(LVA)$^+$ (IB) cells through time. (A) Two binary division events occurred between 26 to 31 hpi, corresponding to cells 3a and b and 5a and b. Scale bar, 10 μm. The 15-min-interval time-lapse videos for each inclusion can be found as Movies S3 and S4 in the supplemental material.

**Steady state versus stem cell population.** The asymmetric production model predicts that after amplification, the RBs act as a stem cell population, producing IBs while self-renewing after every division (Movie S1). In contrast, the direct conversion model predicts that RBs convert directly into IBs and are subsequently replaced by a dividing RB population, producing a steady state of RBs (Movie S2). To determine which of these phenotypes the RB is exhibiting, live-cell imaging at high resolution (40× objective) of cells infected with L2-BMELVA was used to follow RB behavior. Cells infected with L2-BMELVA were imaged for *euo*prom and *hctB*prom expression every 15 min starting at 24 hpi until 60 hpi. We imaged from 24 hpi until 60 hpi, as this covers the end of the RB amplification stage until the end of the cycle. Imaging revealed that the number of RBs (*euo*prom$^+$) per inclusion remained roughly the same throughout the experiment (Fig. 4A and Movie S3). The RBs in each inclusion were easily tracked from one time point to the next and did not disappear when new RBs appeared. We also observed two binary divisions (RB amplification events) occurring between 26 hpi and 31 hpi (Fig. 4A, cells 3a and b and 5a and b, and Movie S3). These newly formed RBs also remained trackable within the inclusion for the remainder of the experiment. These observations are consistent with stem cell-like behavior of the RBs after amplification.

Both models predicted that the IB cell type is maintained at steady state, i.e., IBs mature into EBs, and new IBs replace the maturing cells. To investigate the dynamics of the IB

**FIG 3** Legend (Continued)

IBs, blue; EBs, pink. (B) Simulated kinetics of the direct conversion model if conversion outcompetes replication (panel 1) or replication outcompetes conversion (panel 2). (C) Individual traces of simulated RB and EB kinetics on a per-inclusion level for the asymmetric production and direct conversion models. Colors of individual inclusion traces are paired between the RB and EB cell forms per model simulation. Infections were simulated from 0 to 50 hpi. For panels A and B, the average of 20 simulations/model are shown; cloud represents SEM. Model parameters can be found in Text S1.

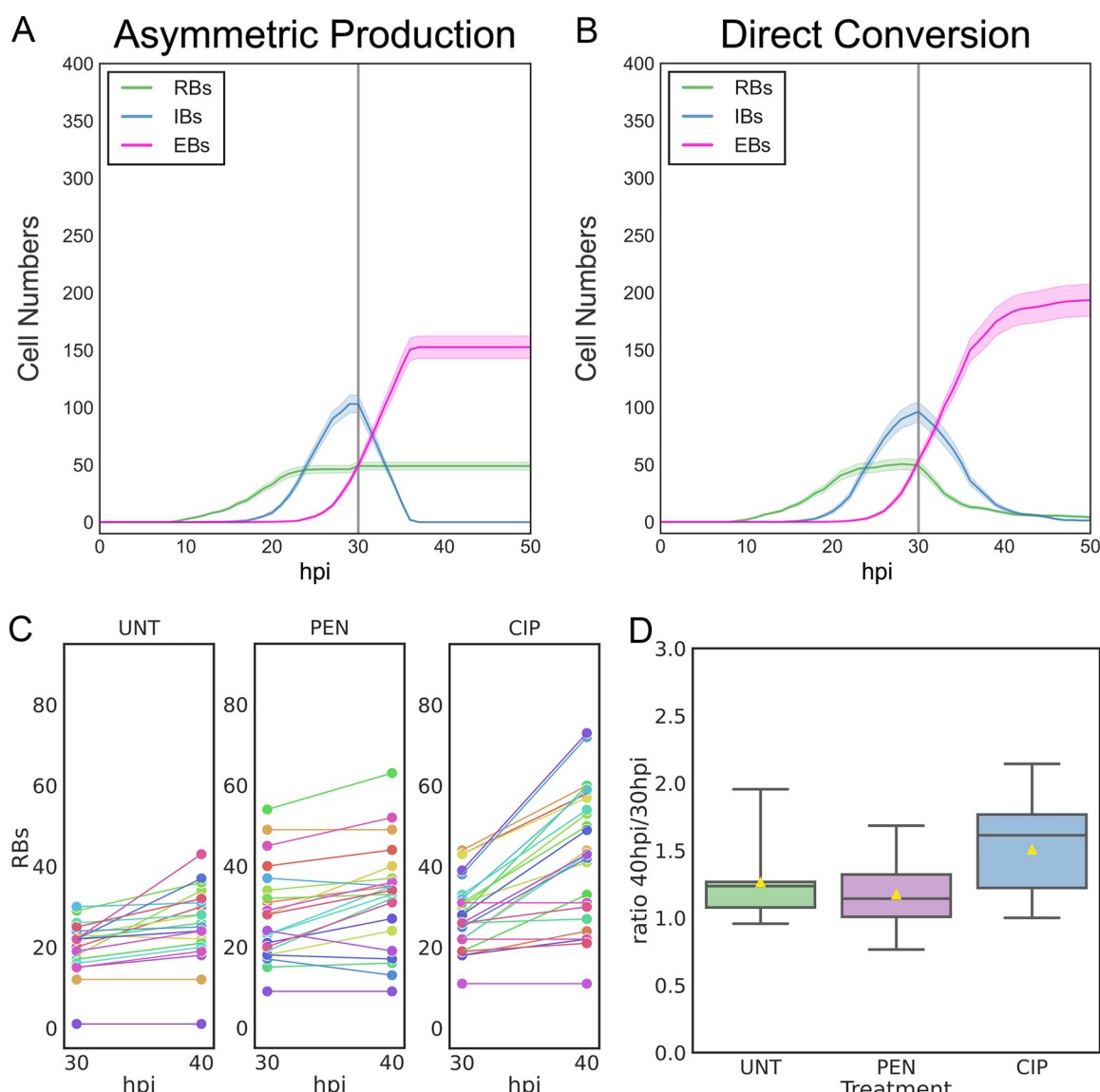

**FIG 5** RBs do not convert into IBs after cell division inhibition. (A and B) Simulated cell form kinetics of cell division inhibition in the asymmetric production (A) and direct conversion model (B). RBs, green; IBs, blue; EBs, pink. Infections were simulated from 0 to 50 hpi. The gray vertical line indicates time of simulated cell division inhibition (30 hpi). Average cell form subpopulation numbers of 20 simulations per model are shown; cloud represents SEM. (C) Cos-7 cells were infected with purified L2-BMALVA EBs. Infected cells were treated at 30 hpi with either vehicle (UNT), penicillin-G (PEN), or ciprofloxacin (CIP). The number of *euo*prom-mNG(LVA)+ cells were counted per inclusion at 30 and 40 hpi. Horizontal lines connect the same imaged inclusions. (D) Boxplot demonstrating the 40:30-hpi ratio of RBs per treatment. The center gray line represents the median number of RBs, and the yellow triangle represents the mean.

population, the L2-BMALVA strain was imaged from 24 hpi until 60 hpi at 15-min intervals. The dynamics of the IBs (*hctA*prom+ cells) resembled steady-state kinetics; *hctA*prom+ cells would disappear while new *hctA*prom+ cells would appear (Fig. 4B; Movie S4).

Together, these data strongly support the asymmetric production model that predicts that RB numbers are amplified between 12 hpi and 28 hpi followed by a stem cell-like behavior, producing one IB at division. In contrast, the IBs behave as a steady-state population where, once formed, the IBs are converted directly into EBs.

**The role of cell division in the RB population.** Simulations of the two competing ABMs predicted different developmental outcomes if cell division was inhibited. The asymmetric production model predicted that RBs produce IBs only at division. Therefore, an immediate block in the formation of new IBs would occur, but RB numbers would remain unaffected if replication was inhibited at 30 hpi (Fig. 5A). In contrast, the direct conversion

model predicted that RB numbers would drop over time if cell division was inhibited at 30 hpi, eventually disappearing as the RBs converted into IBs but were not replaced by further RB replication (Fig. 5B). To test these predictions experimentally, two cell replication inhibitors were used, penicillin (Pen) and ciprofloxacin (Cip). *Chlamydia* does not contain a peptidoglycan cell wall and instead uses peptidoglycan in septum formation; therefore, Pen treatment of *Chlamydia* inhibits cell division (17). Ciprofloxacin prevents bacterial DNA replication by inhibiting topoisomerases and DNA gyrase (18). Cells were infected with L2-BMELVA, and >20 inclusions per treatment were imaged using a 60× 1.4 numerical aperture (NA) objective. Individual inclusions were imaged at multiple Z planes to visualize and quantify all the RBs in the inclusion. Images were collected from the live cultures at antibiotic treatment (30 hpi) and 10 h later (40 hpi) (Fig. 5C and D). The number of *euo*prom$^+$ cells (RBs) was quantified on a per-inclusion basis. The same inclusions were quantified at each time point. Consistent with the confocal time-series experiment, there was a large variation in the number of RBs in individual inclusions, ranging from a single RB to greater than 50 RBs (Fig. 5C). However, the number of RBs per inclusion remained essentially constant between the 30-hpi and 40-hpi time points (Fig. 5C). The mean ratio of RBs at 30 hpi and 40 hpi per inclusion was 1.27 ± 0.28 for untreated, 1.18 ± 0.24 for Pen treated, and 1.51 ± 0.35 for Cip treated (Fig. 5D). There was an increase in RB numbers in each inclusion when treated with Cip, but never more than double. We speculate that although Cip treatment inhibited initiation of DNA replication, it did not significantly impact the continuation of DNA replication and allowed cell division to finish in RBs that had already initiated DNA synthesis. This is in agreement with our published data demonstrating that RBs are undergoing continuous DNA replication, as indicated by a replication index of 1.5 (19). This index is a ratio of sequencing coverage near the origin of replication versus coverage near the terminus and represents the growth rate of the population. An RB replication index of 1.5 demonstrates the presence of partially replicated chromosomes (19).

To confirm that chlamydial DNA replication was inhibited by Cip, digital droplet PCR (ddPCR) was performed on L2-BMELVA-infected samples treated with either Cip, Pen, or mock at 30 hpi. Host monolayers were harvested every 4 h from 26 to 54 hpi. As previously reported, genome copy number continued to increase in the Pen-treated samples (10, 20). There was, however, a large reduction in genome copy accumulation after Cip treatment compared to the mock and Pen-treated samples (Fig. S2B).

**The role of cell division in IB population and EB production.** Simulations of the two competing ABMs also predicted different developmental outcomes in the IB population if cell division was inhibited. The asymmetric production model produces IBs only at division; therefore, blocking cell division at 30 hpi predicted an immediate block in the formation of new IBs, leading to an immediate halt in the increase in IB gene expression (Fig. 6A, asymmetric production). In contrast, after replication was inhibited, the direct conversion model predicted that IB gene expression would continue to increase over a 12-h period as the RBs converted into IBs, at which point IB gene expression would halt, as IBs could not be replenished by RB cell division (Fig. 6A, direct conversion).

To test these predictions experimentally, cell replication was inhibited with Pen, Cip, or genetically by overexpression of FtsI. Penicillin-binding proteins (PBPs) are a suite of enzymes involved in peptidoglycan synthesis and cell division (21). Both PBP2 and PBP3 are (D,D)-transpeptidases that mediate the formation of peptidoglycan. In *Escherichia coli*, PBP2 is required for cell shape and stability, whereas PBP3 is involved in cell division (22). When overexpressed in *E. coli*, PBP2 and PBP3 have been shown to induce lysis in dividing cells (23, 24). However, as *Chlamydia* does not contain a peptidoglycan cell wall, PBP2 and PBP3 have both been co-opted for use in septation (25). Therefore, we overexpressed FtsI (PBP3) to target dividing chlamydial cells. The open reading frame (ORF) of chlamydial *ftsI* was cloned into our translational expression system and tagged with a C-terminal 3×FLAG epitope (12). The *euo*prom-mNG(LVA)_*hctB*prom-mKate2 promoter-reporter cassette was cloned into the E-*ftsI*-3XFLAG vector and transformed into *Chlamydia trachomatis* L2, creating the strain L2-E-*ftsI*-BMELVA. Ectopic expression of full-length FtsI-FLAG in *Chlamydia* was verified by Western blotting (Fig. S2A). Ectopic FtsI expression resulted

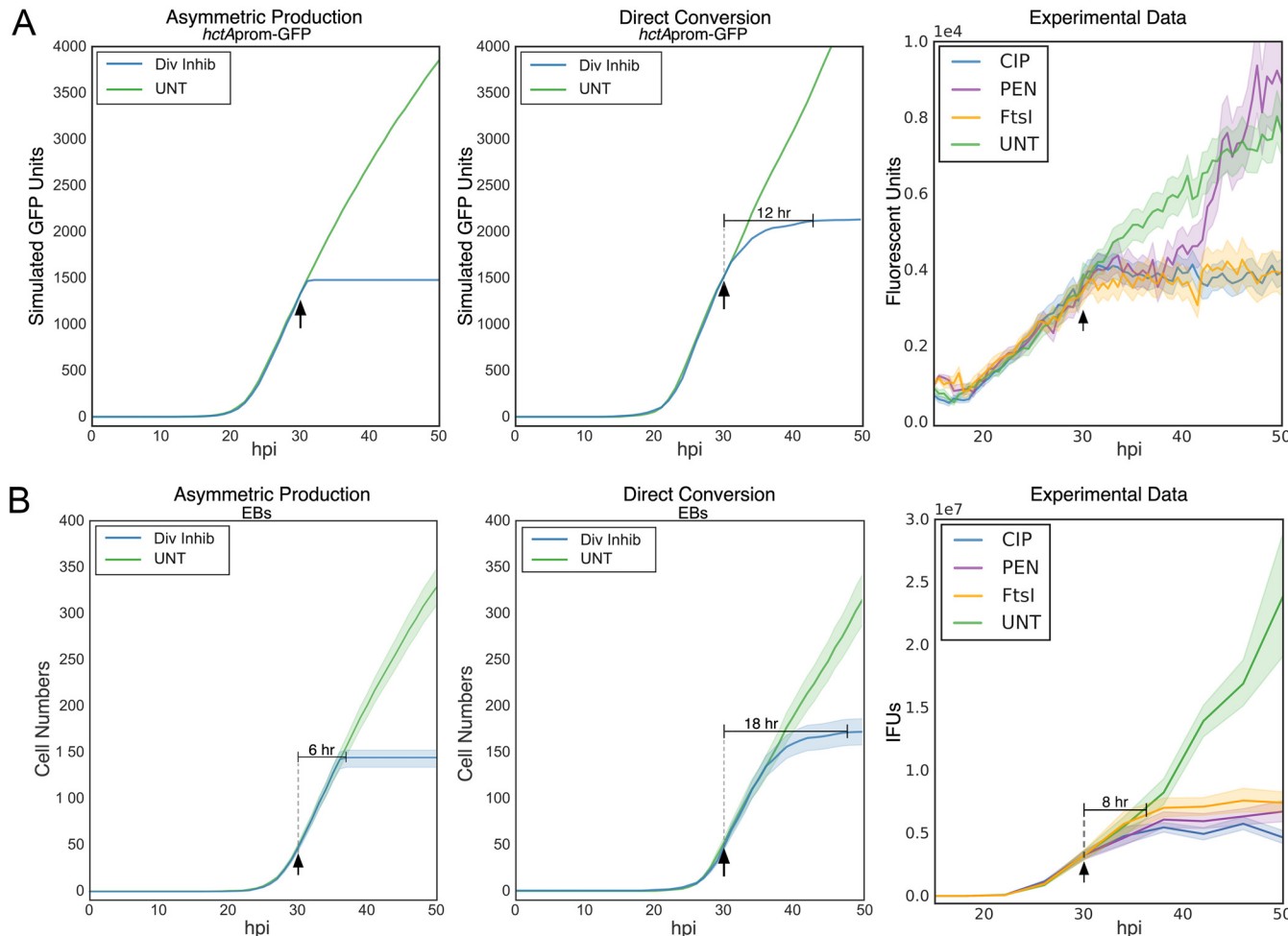

**FIG 6** IB and EB production halts after cell division inhibition. (A) Simulated kinetic outputs of the total accumulation of GFP from the IB-associated promoter *hctA*prom for both the asymmetric production and direct conversion models. (B) Simulated total EB cell numbers for both the asymmetric production and direct conversion models. Untreated, green; cell division inhibition, blue. Infections were simulated from 0 to 50 hpi. Average of 20 simulations per model is shown. (A and B) Experimental data. Cos-7 cells were infected with L2-*ftsI*-3XFLAG-BMAMEO (FtsI induction) or L2-BMAMEO (PEN and CIP). Infected cells were treated at 30 hpi with vehicle (UNT; green), penicillin-G (PEN; purple), ciprofloxacin (CIP; blue), or induced for FtsI (orange). (A) Experimental data. Mean expression kinetics of *hctA*prom-mEos3.2. (B) Experimental data. Mean number of infectious progeny. Arrow indicates time of treatment (30 hpi). Horizontal solid line indicates the time to inhibition of IB or EB formation. Cloud for simulations and fluorescent reporters represents SEM. *n* > 20 inclusions per treatment. Inclusion-forming unit (IFU) cloud represents 95% CI.

in a block in chlamydial replication, as demonstrated by inhibition of chromosome replication measured using digital droplet PCR (Fig. S2B). Additionally, we examined the impact of FtsI overexpression on the RB population using immunofluorescence confocal imaging of host cells infected with L2-E-*ftsI*-BMELVA (Fig. S2C). Infected cells were treated with vehicle only or induced for expression at 20 hpi. Inclusions from the 20-hpi and 30-hpi uninduced samples were filled with bacteria and contained multiple bright *euo*prom-mNG(LVA)⁺ cells, indicating the presence of live RBs capable of active transcription and translation (Fig. S2C). Conversely, inclusions from the 30-hpi FtsI-induced samples were relatively empty compared to the 20-hpi and 30-hpi uninduced samples. A subset of cells in the induced sample stained positively for FLAG (i.e., indicating FtsI induction); these cells were misshapen and contained significantly less *euo*prom-mNG(LVA) fluorescence than uninduced cells, indicating a lack of active expression and suggesting RB lysis had occurred (Fig. S2C). This construct contained a FLAG tag at the C terminus of FtsI, potentially altering its function. We have, however, used C-terminal FLAG tags in other chlamydial protein fusion constructs and not observed any impact on the developmental cycle (12, 19, 26).

An additional dual promoter reporter strain was created using *hctA*prom-mEos3.2 paired with *hctB*prom-mKate2 (L2-BMAMEO). The mEos3.2 GFP variant accumulates over

time, as it does not have the LVA degradation tag and will therefore be present in all cells that have expressed from the *hctA*prom (IBs and EBs). This dual-color promoter reporter cassette was also cloned into the E-*ftsI*-3XFLAG plasmid and transformed into *Chlamydia*, creating L2-E-*ftsI*-BMAMEO. To differentiate between asymmetric production and direct conversion, cells were infected with L2-BMAMEO or L2-E-*ftsI*-BMAMEO and imaged from 10 hpi until 50 hpi (Fig. 6A, experimental data). Pen and Cip were added at 30 hpi to the L2-BMAMEO cultures, and FtsI was induced by adding theophylline (Tph) at 30 hpi to the L2-E-*ftsI*-BMAMEO-infected cells to inhibit chlamydial replication. Live-cell imaging of the *hctA*prom-mEos3.2 produced kinetics consistent with the asymmetric production model simulations. The increase of IB production signal from *hctA*prom was almost immediately inhibited by all three treatments (Fig. 6A, experimental data). As we have previously documented, the *hctA*prom reactivated in the Pen-treated aberrant cells after about an ~8-h delay (10).

The two models also predicted differences in EB production when cell division was inhibited. The asymmetric production model predicted that EB production would halt ~6 h after all treatments indicated (Fig. 6B, asymmetric production). The stochastic direct conversion model simulations predicted that inhibition of cell division would result in a slowing followed by a halt in EB production ~18 h posttreatment (Fig. 6B, direct conversion). To directly test these predictions, cells were infected with L2-BMAMEO or L2-E-*ftsI*-BMAMEO. Pen or Cip were added to L2-BMAMEO cultures, and Tph was added to the L2-E-*ftsI*-BMAMEO-infected cells at 30 hpi. EBs were collected every 4 h from 10 to 50 hpi and used to infect fresh host cells for EB quantification. Our data show that EB production was inhibited ~8 h post-cell division inhibition with all three inhibitors (Fig. 6B, experimental data). These data, taken together, support a division-dependent asymmetric IB production model and not stochastic direct conversion of an RB to an IB.

**EB formation is independent of continued IB production.** Both of our ABMs rely on the direct conversion of IBs into EBs without replication. To test this assumption, we infected cells with L2-BMELVA and used confocal imaging to assay for the expression of the EB-associated promoter, *hctB*prom. Cells were infected with L2-BMELVA, and replication was inhibited by treatment with Pen or Cip at 20 hpi. Untreated and treated samples were fixed and stained for DNA (DAPI) at 20 hpi and 30 hpi and imaged for DAPI, GFP, and RFP fluorescence. Before treatment (20 hpi), all inclusions had *euo*prom+ cells (RBs), DAPI-only positive cells (IBs), and no *hctB*prom+ cells (EBs) (Fig. 7A; Fig. S3A). At 30 hpi, the untreated inclusions contained *euo*prom+ cells (RBs), DAPI-only cells (IBs), and *hctB*prom+ cells (EBs) (Fig. 7B; Fig. S3A). The inclusions treated with Cip for 10 h contained *euo*prom+ cells (RBs) and *hctB*prom+ cells (EBs), but fewer DAPI-only cells (IBs) than untreated (Fig. 7B; Fig. S3B). Pen-treated inclusions contained large aberrant *euo*prom+ cells (aberrant RBs), small *hctB*prom+ cells (EBs), and fewer DAPI-only cells (IBs) at 30 hpi (Fig. 7B; Fig. S3C).

We also asked whether inhibiting cell division through FtsI ectopic expression affected IB-to-EB formation. Cells were infected with L2-E-*ftsI*-BMELVA and induced for FtsI expression at 20 hpi. The infected cells were fixed and stained with DAPI, and confocal images were taken for DAPI, GFP, and RFP signals at 20 and 30 hpi. The inclusions in the L2-E-*ftsI*-BMELVA-infected cells at 20 hpi contained *euo*prom+ cells (RBs) and DAPI-only stained cells (IBs) and little to no *hctB*prom+ cells (EBs) (Fig. S3D; Fig. S4A). The uninduced L2-E-*ftsI*-BMELVA inclusions at 30 hpi contained *euo*prom+ cells (RBs), DAPI-only positive cells (IBs), and a significant number of *hctB*prom+ cells (EBs) (Fig. S4B). However, the Tph-induced L2-E-*ftsI*-BMELVA inclusions at 30 hpi had misshapen large chlamydial cells with just a trace of *euo*prom+ signal (RBs), fewer DAPI-positive cells (IBs), and a significant number of *hctB*prom+ cells (EBs) (Fig. 7B; Fig. S3D). Taken together, these data support the hypothesis that the IB matures directly into the EB cell form without undergoing cell division.

**IBs convert directly to EBs.** To verify that the IB directly matures into the EB without cell division (*hctA*prom+ to *hctA*prom+ or *hctB*prom+), cells were infected with L2-BMAMEO to visualize the *hctA*prom+ and *hctB*prom+ cells, and cell division was inhibited with Cip at 18 hpi. Infected cells were then fixed and stained with DAPI at 22 hpi and 34 hpi. The inclusions were imaged for DNA (DAPI), GFP (*hctA*prom), and RFP (*hctB*prom) (Fig. 8A and B). The expression levels of GFP and RFP were determined for

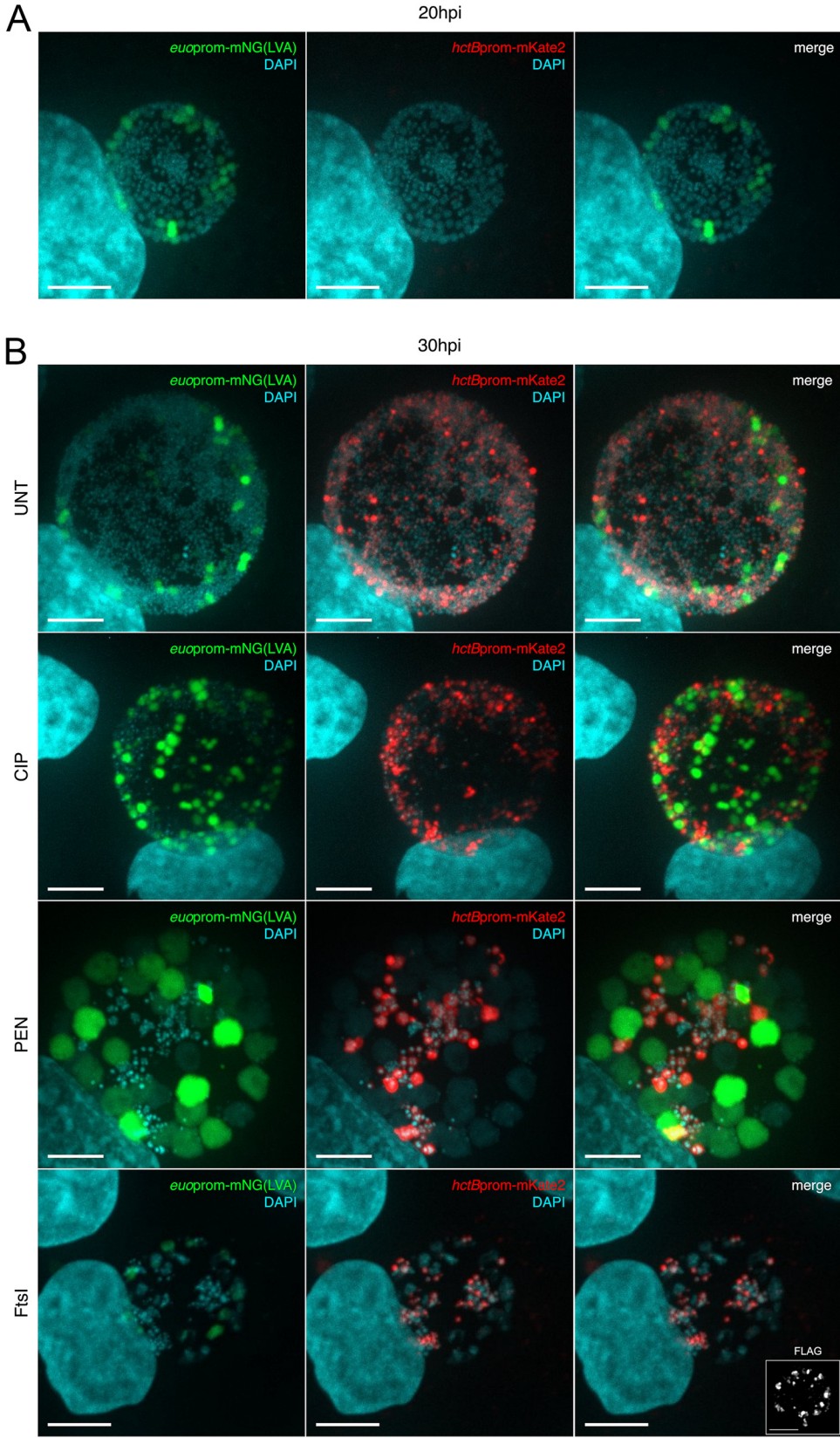

**FIG 7** IB-to-EB development is replication independent. Cos-7 cells were infected with either purified L2-BMELVA or L2-E-*ftsI*-3XFLAG-BMELVA. L2-BMELVA-infected cells were treated with either vehicle (UNT), ciprofloxacin (CIP), or penicillin-G (PEN), and L2-E-*ftsI*-3XFLAG-BMELVA infected cells were induced for FtsI at 20 hpi. Samples were fixed at

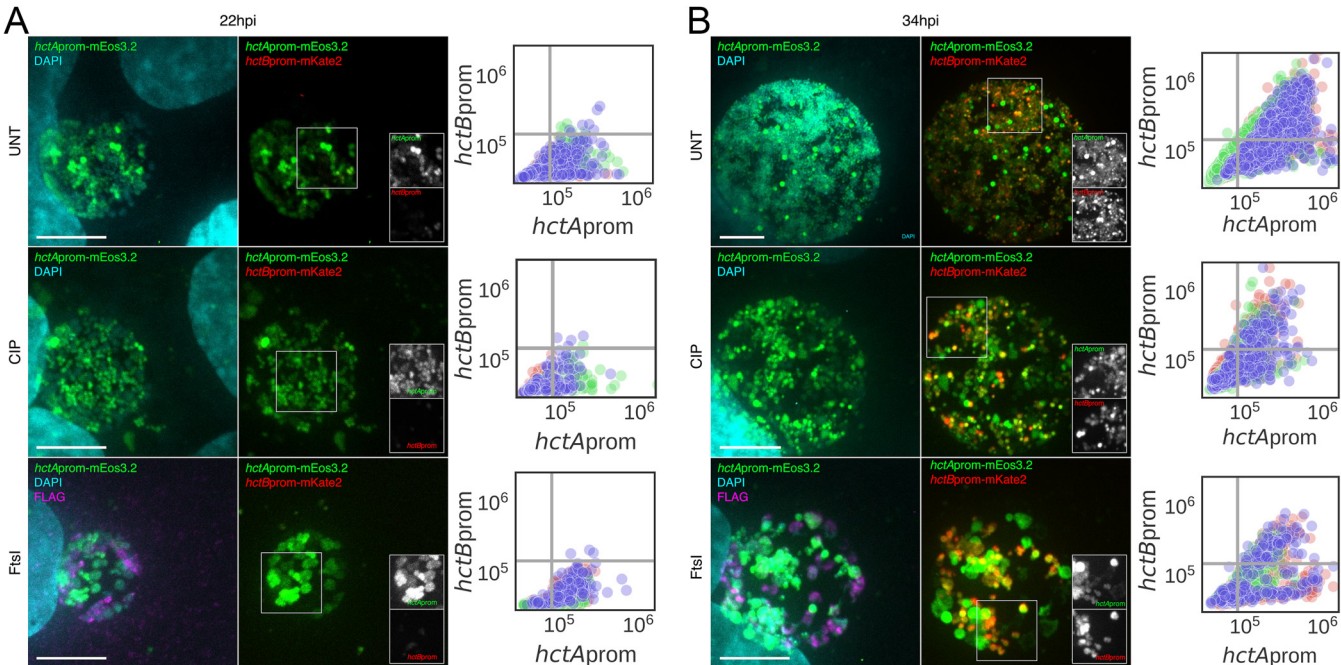

**FIG 8** IBs mature directly into EBs. Cos-7 cells were infected with L2-E-*ftsI*-3XFLAG-BMAMEO. Infected cells were treated at 18 hpi with either vehicle (UNT) or ciprofloxacin (CIP) or induced for FtsI-3XFLAG expression (FtsI). Samples were fixed at 22 hpi or 34 hpi and stained with DAPI and an anti-FLAG antibody. Images are z-projections from confocal micrographs showing *hctA*prom-mEos3.2 (green), *hctB*prom-mKate2 (red), DAPI (cyan). and anti-FLAG (magenta). (A) Representative confocal micrographs of 22-hpi cells along with quantification of *hctA*prom-mEos3.2 and *hctB*prom-mKate2 expression levels at the single-cell level. Quantification of DAPI-positive cells was performed using TrackMate from 3 individual inclusions per treatment per time point. Each color corresponds to chlamydial cells quantified within the same inclusion. (B) Confocal micrographs of L2-E-*ftsI*-3XFLAG-BMAMEO-infected cells at 34 hpi either treated with vehicle (UNT) or CIP or induced to express FtsI. Inserts demonstrate the overlap of *hctA*prom-mEos3.2 and *hctB*prom-mKate2 within single cells. Quantification of DAPI-positive cells was performed using TrackMate from 3 individual inclusions per treatment per time point. Each color corresponds to chlamydial cells quantified within the same inclusion. *n*, > 500 cells per treatment. Scale bar, 10 $\mu$m.

>500 individual chlamydial cells in three representative inclusions by identifying cells using the DAPI channel and measuring expression levels using the TrackMate plugin in Fiji (Fig. 8A and B). Confocal microscopy of untreated cells at 22 hpi revealed that there were a number of DAPI-only (RBs) and *hctA*prom$^+$ cells (IBs) with very few *hctA*prom$^+$ and *hctB*prom$^+$ cells (IB→EBs) (Fig. 8A, UNT). At 34 hpi, there were again populations of DAPI-only (RBs) and *hctA*prom$^+$ cells (IBs); however, many of the *hctA*prom$^+$ cells were also positive for *hctB*prom (IB→EB) (Fig. 8B, UNT). Similar trends were seen when cell division was inhibited by the addition of Cip at 18 hpi. The chlamydial population at 22 hpi consisted of primarily DAPI-only (RBs) and *hctA*prom$^+$ cells (IBs) (Fig. 8A, CIP), while many of the *hctA*prom$^+$ cells were also *hctB*prom$^+$ (IB→EB) by 34 hpi (Fig. 8B, CIP). This increase in the number of double-positive cells in cell division-inhibited *Chlamydia* suggests that the *hctA*prom$^+$ cells were activating *hctB*prom over time.

We also asked whether the IB could directly become an EB after replication was inhibited by ectopic expression of FtsI. Cells were infected with L2-E-*ftsI*-BMAMEO, and FtsI expression was induced at 18 hpi. At 22 hpi, there were a number of DAPI-only (RBs) and *hctA*prom$^+$ cells (IBs) in the induced population. Anti-FLAG staining revealed that many of the DAPI-only cells were expressing FtsI-FLAG (Fig. 8A, FtsI). The inclusions at this time point contained very few *hctB*prom$^+$ cells (EBs). At 34 hpi, there was a significant increase in the number of *hctA*prom/*hctB*prom double-positive chlamydial cells (IB→EB), suggesting

**FIG 7** Legend (Continued)

20 hpi (pretreatment) or 30 hpi and stained with DAPI. Images are z-projected confocal micrographs showing *euo*prom-mNG(LVA) (green), *hctB*prom-mKate2 (red), and DAPI (cyan). (A) Representative confocal micrograph of a 20-hpi inclusion. (B) Representative confocal micrographs of 30-hpi UNT, CIP, PEN and FtsI-induced infections. Insert demonstrates positive anti-FLAG staining in the FtsI-induced sample. Scale bar, 10 $\mu$m. See Fig. S3 in the supplemental material for chlamydial cell quantification and Fig. S4 for the uninduced FtsI sample.

the *hctA*prom$^+$ cells activated *hctB*prom without cell division (Fig. 8B, FtsI). The uninduced samples showed a similar pattern. These data together demonstrate that IBs (*hctA*prom$^+$) mature directly into EBs (*hctB*prom$^+$) and that cell division is not required for this progression.

## DISCUSSION

The developmental cycle of *Chlamydia* is central to its ability to cause disease. The cycle produces phenotypically distinct cell types with functional specificity. This includes the RB cell, which replicates, leading to organism proliferation, and the EB cell type, which mediates entry into new cells, disseminating the infection to new hosts. The developmental cycle has conventionally been broken down into two stages, RB replication and EB conversion. While the broad strokes of the overall cycle are described, the molecular details that regulate this process are poorly understood. Our data suggest that the current understanding of the cycle is an oversimplification. We propose that the developmental cycle can be divided into multiple stages, including EB germination, RB amplification and maturation, IB production, and EB formation.

Our data show that the chlamydial developmental cycle produces at least three different phenotypic cell types, RB (*euo*prom$^+$), IB (*hctA*prom$^+$), and EB (*hctB*prom$^+$). The data also demonstrate that RB proliferation initiates at ~10 to 12 hpi and reaches a plateau by 24 to 26 hpi, after which RB numbers remain virtually unchanged until the end of the cycle. Remarkably, at the individual inclusion level, the number of RBs at plateau was highly variable, with some inclusions containing just a few RBs, while others had as many as 60. To understand the potential mechanisms that underlie these observations, we developed computational agent-based models (ABMs). Two mechanistic models were developed that could recapitulate our experimental data. Both models estimate the germination time based on published time to first RB replication (16) and time to *euo*prom expression (this study). The models differ primarily in the IB production mechanism, which we hypothesize is the committed step to EB formation. Regulatory mechanisms, such as RB access to or competition for inclusion membrane contact (26), reduction in RB size (7), and responses to changes in nutrient availability (27), have been proposed to explain the regulation of EB formation. Although the triggering signal for differentiation differs in these models, all propose a stochastic direct RB-to-EB conversion mechanism. Therefore, the direct conversion model utilized a direct conversion mechanism to control RB amplification and IB production. Our second ABM, the asymmetric production model, used an RB maturation/asymmetric division mechanism to control RB amplification and IB production. In this model, we proposed two RB subtypes, the RB$_R$, which, upon replication, produces two identical RB$_R$ daughter cells, and an RB$_E$ that, upon cell division, produces one IB and one RB$_E$. The RB$_R$ subtype matures into the RB$_E$ subtype during the first 24 h of infection. For both models, the IB cell exits the cell cycle and matures directly into the EB. Although both models could reproduce the measured kinetics of the developmental cycle at the population level, only the maturation/asymmetric model could reliably reproduce the observed kinetics at the single-inclusion level. Even after constraining replication and conversion to avoid RB extinction or overpopulation, the stochastic direct conversion model resulted in runs of over- and under-IB production that were not readily evident in the measured data.

The asymmetric production model predicted that the mature RB cell (RB$_E$) acts as a stem cell, producing one IB upon division, while the other daughter cell remains an RB$_E$. The direct conversion model predicted that RB numbers are at steady state where cell division creates new RBs, while other RBs convert into IBs. Using live-cell microscopy, we determined that individual RBs were consistently present and trackable from time point to time point over a 30-h time period, strongly suggesting the RB population is acting like stem cells and not a steady-state population. Conversely, both models predicted that the IB population is at steady state, with IBs forming and transitioning into EBs, while new IBs replace those that became EBs. The behavior of individual IBs revealed by live-cell imaging was consistent with a steady-state population, with IBs disappearing and new ones appearing over time.

The models also predicted that inhibition of cell replication would have differing effects on the RB, IB, and EB populations. The stochastic direct conversion model predicted that RB numbers would decline after cell division inhibition, as RBs that directly converted to IBs were not replaced by new RBs through cell division. In contrast, the asymmetric production model predicted that inhibition of cell division would have no impact on the RB population. We found that treatment with two different cell replication inhibitors (Pen and Cip) resulted in no decrease in the RB population over a 10-h period, again strongly supporting an asymmetric division model.

The kinetics of the IB and EB population was also predicted to differ in the two models after replication was inhibited. The direct conversion model predicted that inhibition of cell division would lead to a protracted inhibition of new IB formation, resulting in decreasing IB production over ~12 h, while EB formation would also slowly decline until completely inhibited at ~18 h after treatment. The asymmetric production model predicted that inhibiting cell division would inhibit new IB production nearly immediately and inhibit EB formation after the ~8-h IB-to-EB maturation time (10). Live-cell microscopy demonstrated that the kinetics of cell division inhibited *Chlamydia*, which clearly supported the asymmetric production model over the direct conversion model, as IB production halted nearly instantly, and EB production halted after ~8 h.

Both models predicted that IBs mature directly into EBs. This prediction is supported by the data showing that the IB cell type progressed to *hctB*prom expression after inhibiting cell division with Cip or Pen or through ectopic expression of FtsI. Additionally, we demonstrated that IBs mature directly from *hctA*prom single-positive to *hctA*prom/*hctB*prom double-positive cells, even after inhibiting cell division with Cip or overexpression of FtsI.

Cell size has been proposed as a mechanism controlling stochastic RB-to-IB conversion. The average chlamydial cell size becomes smaller and more heterogeneous over time (7). Our asymmetric production model also predicts that cell size will behave similarly, decreasing in size and increasing in heterogeneity over time. In our model, early in the developmental cycle, the $RB_R$ cell progresses through the cell cycle and divides to produce two $RB_R$s, which immediately reenter the cell cycle, gaining mass before the next division. As the cycle progresses, $RB_R$s mature into $RB_E$s. The $RB_E$ continues in the cell cycle and, upon division, produces one $RB_E$ and one IB. After division, the $RB_E$ remains in the cell cycle, increasing in size; however, the IB does not reenter the cell cycle, does not begin to add mass, and is therefore smaller than the $RB_{R/E}$ cells. The IB then proceeds to reduce in volume as it matures to the EB cell over an ~8-h time frame (10). This process would produce IB cells of various sizes with a trend toward smaller and smaller cells in the inclusion, as early IBs increase in number compared to RBs.

Overall, the data support a developmental cycle that includes an asymmetrically dividing RB population. Asymmetry has been documented for both the EB and RB cells (28–31). In addition, it has been shown that the division plane can form asymmetrically during RB division (28, 32). Asymmetric cell division is a common mechanism to generate phenotypically distinct cell populations in bacteria. Many of these systems have evolved to create two-cell populations; one cell acts as the stem cell, while the other cell disseminates the bacterial colony to new environments. Both *Caulobacter crescentus* and some members of *Chlamydia's* nearest phylogenetic neighbors, the *Planctomycetes*, undergo a division cycle that includes a surface-attached mother cell that produces a planktonic swarmer cell upon division in order to extend the population to new ecological niches (33, 34). The swarmer cell in the case of *C. crescentus* is nonreplicating and is out of the cell cycle (35). Our data support a similar role for the EB cell, as the EB disseminates the infection, does not replicate, and is out of the cell cycle (19).

The mother/swarmer cell developmental model fits our data for IB and EB production but does not explain RB expansion. We have modeled this process (RB expansion to RB and IB asymmetric division) as maturation over time from an RB that produces two RBs, thereby increasing the dividing cell population ($RB_R$) to an RB mother/stem cell ($RB_E$) that produces an IB through asymmetric division. Currently, the mechanism for $RB_R$ to $RB_E$ maturation is unknown but could be influenced by several factors, such as EB age at infection, differences

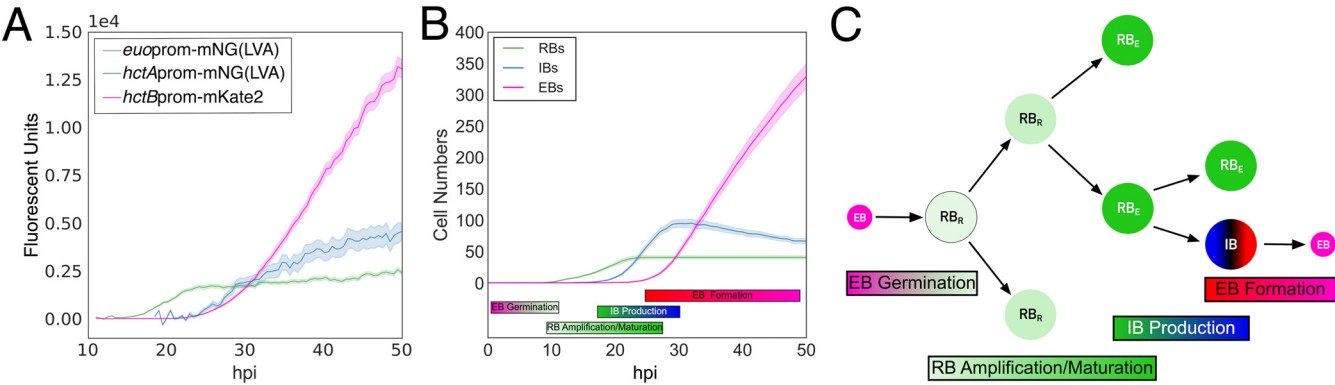

**FIG 9** Model of the developmental cycle. (A) Cell form-specific live-cell kinetics from Fig. 1. (B) Simulated developmental cycle using the asymmetric division/maturation model. (C) Schematic of the developmental model. Model consists of EB-to-RB germination, RB amplification/maturation, asymmetric IB production, and direct IB-to-EB formation. RBs (*euo*), green; IB (*hctA*), blue→black→red; EB (*hctB*), pink.

in nutrient acquisition, and the effects of inclusion membrane interaction, or through a yet-to-be-described stochastic maturation mechanism. Additionally, our data show that this expansion step produces significant heterogeneity in RB$_E$ numbers in individual inclusions. This differential amplification is potentially a novel evolutionary adaptation to balance RB maturation and early EB production, i.e., late maturation leads to late EB production but a high EB production rate, whereas early maturation leads to early EB production with a low EB production rate. We currently have no expression markers that show differences between the two theoretical RB subtypes, the RB$_R$ and the RB$_E$. Our data and subsequent model suggest that this is the best explanation for our data and will be a continuing area of investigation.

The asymmetric model suggests that asymmetry is generated during cell division after a maturation process. We have not, however, detected differential expression between forming daughter cells during division using our currently available dual-color promoter reporters. The two forming daughter cells appear to express similar levels of fluorescence from the *euo*prom-mNG(LVA) reporter and also appear to be similar sizes. This has been reported by other studies that showed that although there is evidence that the division plane initiates asymmetrically, at cytokinesis, the two daughter cells appear to be of similar size (8, 18, 24, 25). Our data suggest that differential gene regulation of the *hctA* promoter in the two daughter cells happens after the cells have separated, thereby increasing the difficulty in identifying the daughter cell pairs.

Our data support a four-stage model, EB germination, RB amplification and maturation, IB production, and EB formation (Fig. 9). A key aspect of this model is the stochastic amplification and maturation of RBs (RB$_R$) from an expanding population to a stem cell RB population (RB$_E$) that produces IBs through asymmetric replication reminiscent of stalk/swarmer cell dynamics (Fig. 9). This is followed by an EB formation stage where the IB undergoes a dramatic phenotypic change that includes the expression of the nucleoid-associated protein, HctA, and ultimately ends with the expression of a second nucleoid-associated protein, HctB. This EB formation stage takes ~8 to 10 h, ultimately resulting in the infectious EB (10) (Fig. 9). Our model suggests that the cycle starts with RB amplification and quickly converts to a mixed environment of RB$_R$s producing RBs and RB$_E$s producing IBs. The IB phenotype takes ~8 to 10 h to complete maturation into the infectious EB. Our data suggest that replication is significantly faster than IB-to-EB maturation; therefore, IBs accumulate and are a significant cell type in the inclusion for most of the cycle, outnumbering RBs as soon as ~18 hpi. It is currently unclear what role the IB cell type has in the cell biology of chlamydial infection.

Temporal gene expression during chlamydial infection has been extensively reported at the population level. These data have historically categorized developmental gene expression into early-, mid-, and late-expressed genes (8, 36). However, these studies were all performed at a population level and normalized accordingly. Our model and supporting

data suggest that this gene expression cascade is explained by a shift in cell types within the inclusion. Early in the cycle ($\sim$10 to 16 hpi), the inclusion is dominated by RB cells; by 18 hpi, there is a mix of RBs and IBs; and by 24 hpi, early IBs have matured into EBs, and IBs begin to outnumber the RBs. Additionally, from 30 hpi and on, EBs quickly accumulate, becoming the dominant cell form, overwhelming the gene expression signal of the other cell types. Our data suggest that to fully understand the developmental cycle, the kinetically dynamic mixed-cell environment needs to be taken into consideration.

Our experiments and models have focused on *Chlamydia* serovar L2 living in close to ideal growth conditions (10) However, the developmental cycle is shared by all of the *Chlamydiaceae* family. Our studies focused on serovar L2, as it has become established as a model organism for understanding the basic life cycle of these organisms, as it lends itself to genetic manipulation and infects cultured cells efficiently. Our studies are likely to result in a further understanding of the chlamydial developmental cycle across the family. Eventually, we would like to determine if what is reported here describes all the *Chlamydia* species and chlamydial-like organisms, but further studies are needed. Additionally, it is likely that *Chlamydia* reacts and adapts the cycle to nutrient and other stresses. The creation of an ABM that models the mechanisms of the developmental cycle under ideal conditions will provide a tool to visualize and understand the convolved data obtained from nutrient-limiting, pharmacological, genetic, and molecular experiments. Ultimately, a better mechanistic understanding of the developmental cycle will lead to novel therapeutics targeting development, as breaking the cycle will eliminate dissemination and chlamydial disease.

## MATERIALS AND METHODS

**Organisms and cell culture.** Cos-7 cells were obtained from ATCC. Cells were maintained in a 5% $CO_2$ incubator at 37°C in RPMI 1640 (Cellgro) supplemented with 10% fetal plex (FP) and 10 mg/mL gentamicin. All *C. trachomatis* L2-bu434 (L2) strains were harvested from Cos-7 cells. Elementary bodies were purified by density centrifugation using 30% MD-76R at 48 h postinfection (37). Purified elementary bodies were stored at −80°C in sucrose-phosphate-glutamate buffer (SPG) (10 mM sodium phosphate [8 mM $K_2HPO_4$, 2 mM $KH_2PO_4$], 220 mM sucrose, and 0.50 mM L-glutamic acid, pH 7.4). *Escherichia coli* ER2925 ($dam^-$ $dcm^-$) was utilized to produce unmethylated plasmids for transformation into *Chlamydia*.

**Promoter reporter and inducible expression constructs.** All constructs were created in the p2TK2SW2 plasmid background (38). Promoters and the *ftsI* ORF were amplified from *C. trachomatis* L2 (LGV Bu434) genomic DNA using the indicated primers (see Table S1 in the supplemental material). Fluorescent reporters were ordered as gBlocks and cloned using the In-Fusion HD EcoDry Cloning kit (TaKaRa). Promoter reporter constructs were created as previously described (10, 39). The p2TK2SW2-E-*ftsI*3XFLAG was generated by replacing the *Clover* gene with the *ftsI* ORF in the previously created p2TK2SW2-E-*Clover*-3XFLAG plasmid (12). Dual promoter reporter cassettes [*euo*prom-mNG(LVA)_*hctB*prom-mKate2 and *hctA*prom-mEos3.2_*hctB*prom-mKate2] were then inserted upstream of E-*ftsI*-3XFLAG to produce the p2TK2SW2-E-*ftsI*-3XFLAG_*euo*prom-mNG(LVA)_*hctB*prom-mKate2, and p2TK2SW2-E-*ftsI*-3XFLAG_*hctA*prom-mEos3.2_*hctB*prom-mKate2 constructs.

**Chlamydial transformation and isolation.** Transformation of *C. trachomatis* L2 was performed as previously described with selection using 500 ng/$\mu$L spectinomycin (38). Clonal isolation of transformants was achieved by inclusion isolation (MOI < 1) via micromanipulation. To confirm clonality, each construct was purified from the chlamydial transformants and transformed into *E. coli*, and five colonies were sequenced. The resulting strains are summarized in Table 1.

**Infections.** Infections were synchronized by incubating Cos-7 cells with *C. trachomatis* L2 EBs in Hanks' balanced salt solution (HBSS) (Gibco) for 15 min at 37°C while rocking. The inoculum was removed, and cells were washed with prewarmed (37°C) HBSS with 1 mg/mL heparin. The HBSS was replaced with fresh RPMI 1640 containing 10% FP, 10 $\mu$g/mL gentamicin, 1 $\mu$g/mL cycloheximide, and 1 mg/mL heparin sodium. Chlamydial cell division was inhibited by the addition of 0.5 $\mu$g/mL ciprofloxacin or 1 U/mL penicillin-G to the media. Expression of *ftsI*-3XFLAG was induced by the addition of 0.5 mM theophylline to the media (12).

**Replating assays.** EBs were isolated from infected Cos-7 cells by scraping the host monolayer followed by centrifugation at 4°C for 30 min at 18,213 relative centrifugal force (rcf). EB pellets were resuspended in 4°C RPMI via sonication and used to infect Cos-7 cells in polystyrene 96-well microplates in a 2-fold dilution series. Infected cells were incubated for 29 h followed by methanol fixation. Fixed cells were stained with 4′,6-diamidino-2-phenylindole (DAPI) for visualization of host-cell nuclei and anti-mitochondrial outer membrane permeabilization (MOMP) antibody conjugated to fluorescein isothiocyanate (FITC; Thermo Scientific) for visualization of *Chlamydia*. Monolayers were imaged with an Andor Zyla scientific complementary metal oxide semiconductor (sCMOS), Nikon Eclipse TE300 inverted microscope, the scopeLED illumination system at 470 nm and 390 nm, and BrightLine band-pass emissions filters at 514/30 nm and 434/17 nm. Automated image acquisition was performed using $\mu$Manager software (40). Inclusion numbers were quantified with custom scripts in ImageJ and analyzed in custom Python notebooks as previously described (10, 37, 39).

**Genome number quantification.** Total DNA was isolated from infected Cos-7 cells during active infections using an Invitrogen PureLink genomic DNA minikit. A QX200 digital droplet system (Bio-Rad)

was utilized for quantification of chlamydial genomic copies. A 2× ddPCR supermix for probes-no dUTP kit (Bio-Rad) and a custom *copN*-specific primer/probe set was used for DNA detection (Table S1).

**Live-cell microscopy.** Monolayers were seeded on a multiwell glass-bottom plate and infected with *Ctr*-L2 EBs. Infections were grown in an Oko-Touch $CO_2$-heated stage incubator. Fluorescence images were acquired via epifluorescence microscopy using a Nikon Eclipse TE300 inverted microscope with a ScopeLED lamp at 470 nm and 595 nm and BrightLine band-pass filters at 514/30 nm and 590/20 nm. We used 20×/0.4 NA dry, 40×/0.6 NA dry, and 60×/1.40 NA oil objective lenses. Differential interference contrast (DIC) was used to autofocus images. Image acquisition was performed using an Andor Zyla sCMOS camera in conjugation with $\mu$Manager software (40). Images were taken in 30-min intervals unless otherwise stated. Imaging ranged from 10 to 60 h after *Ctr*-L2 infection, depending on the experiment. Multiple fields were imaged for each treatment, and the fluorescent intensity of individual inclusions was monitored using the TrackMate plugin in Fiji (14). Inclusion fluorescent intensities were averaged and graphed in Python as previously described (39).

**Confocal microscopy.** Cos-7 cells were seeded onto glass coverslips and infected with the appropriate *Ctr*-L2 strains. Samples were fixed at the designated times in 2% paraformaldehyde in phosphate-buffered saline (PBS) at room temperature overnight. Samples were then washed with filtered PBS and stained with DAPI to visualize DNA and monoclonal anti-FLAG M2 antibody (Sigma, Thermo Scientific) with Alexa 647 anti-mouse secondary antibody to visualize *FtsI*-3XFLAG expression. Coverslips were mounted onto a microscope slide using Mowiol (100 mg/mL 150 Mowiol 4-88, 25% glycerol, 0.1 M Tris, pH 8.5). Images were acquired using a Nikon spinning disk confocal inverted microscope with a 100×/1.45 NA oil objective with a laser lamp at 405 nm, 490 nm, 568 nm, and 660 nm. Image acquisition was performed using an Andor Ixon electron-multiplying charge-coupled-device (EMCCD) camera and the Nikon Elements software. Collected images were taken as z stacks at 0.2-$\mu$m intervals, capturing the entirety of each inclusion. Multiple inclusions were imaged for each treatment and time point, and quantification of individual cells was performed using TrackMate. Chlamydial cell numbers were then analyzed in custom Python notebooks. Representative confocal micrographs are maximal-intensity projections of 3D data sets, and brightness and contrast were adjusted equally for comparisons.

**Western blot analysis.** To ensure that ectopically expressed FtsI was full size, infected monolayers were lysed in reducing lane marker sample buffer, and protein lysates were separated on a 6% SDS-PAGE gels and transferred to a nitrocellulose membrane for Western blot analysis of the FLAG-tagged protein. The membrane was blocked with PBS with 0.1% Tween 20 (PBS-T) and 5% nonfat milk prior to incubating in monoclonal anti-FLAG M2 antibody (1:40,000; Sigma, Thermo Scientific) overnight at 4°C followed by goat-anti mouse IgG-horseradish peroxidase (HRP) secondary antibody (Invitrogen) at room temperature for 2 h. The membrane was developed with the SuperSignal West Dura luminol and peroxide solution (Thermo Scientific) and imaged using an Amersham Imager 600.

**Computational modeling.** Modeling was done using the CellModeller platform (15). The model description scripts and model data analysis scripts are described in Text S1 and available on GitHub (https://github.com/SGrasshopper/Chlamydial-developmental-cycle).

**Data availability.** All data, bacterial strains, and methodologies are available upon request.

## SUPPLEMENTAL MATERIAL

Supplemental material is available online only.
**VIDEO S1**, MOV file, 6.8 MB.
**VIDEO S2**, MOV file, 6.8 MB.
**VIDEO S3**, MOV file, 3.8 MB.
**VIDEO S4**, MOV file, 4.4 MB.
**TEXT S1**, PDF file, 0.4 MB.
**FIG S1**, TIF file, 0.4 MB.
**FIG S2**, PDF file, 1.7 MB.
**FIG S3**, TIF file, 1.1 MB.
**FIG S4**, TIF file, 7.3 MB.
**TABLE S1**, DOCX file, 0.3 MB.

## ACKNOWLEDGMENTS

This work was supported by National Institutes of Health grants R01AI130072, R21AI135691, and R21AI113617 (N.A.G., S.S.G.). Additional support was provided by the Institute for Modeling Collaboration and Innovation (IMCI) Data Access Grant from the University of Idaho and a pilot grant (NIH COBRE P20GM104420).

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
