## [Reviewer comments · mSystems]

Computational Modeling of the Chlamydial Developmental Cycle Reveals a Potential Role for Asymmetric Division

Travis Chiarelli, Nicole Grieshaber, Cody Appa, and Scott Grieshaber

Corresponding Author(s): Scott Grieshaber, University of Idaho

Review Timeline:

Submission Date:	January 16, 2023
Editorial Decision:	February 13, 2023
Revision Received:	February 14, 2023
Accepted:	February 15, 2023

Editor: Neha Garg

Reviewer(s): The reviewers have opted to remain anonymous.

Transaction Report:

DOI: <https://doi.org/10.1128/msystems.00053-23>

February 13, 2023

Dr. Scott S Grieshaber
University of Idaho
Biological Sciences
Moscow, ID 83844

Re: mSystems00053-23 (**Computational Modeling of the Chlamydial Developmental Cycle Reveals a Potential Role for Asymmetric Division**)

Dear Dr. Scott S Grieshaber:

Thank you for submitting your manuscript to mSystems. We have completed our review and I am pleased to inform you that, in principle, we expect to accept it for publication in mSystems. However, acceptance will not be final until you have adequately addressed the reviewer comments. These are minor modifications, which can be returned within few days.

Preparing Revision Guidelines

Sincerely,

Neha Garg

Editor, mSystems

Journals Department
Reviewer comments:

Reviewer #1 (Comments for the Author):

The authors have provided additional data and/or clarifications addressing concerns regarding original submissions. Two minor issues remain that should be addressed.

1)The authors demonstrate that the overexpression of FtsI with a C-terminal FLAG tag blocks DNA replication. Text should include an additional statement indicating that the observed effect could be due to overexpression, the location of the tag, and/or the highly charged nature of the FLAG tag.

2)The location of the boxed region in the merged image following FtsI induction in Fig. S2 does not match the zoomed region in the figure.

Reviewer #2 (Comments for the Author):

The authors have addressed all comments and questions raised from the previous submission. No additional concerns are noted.

February 15, 2023

Dr. Scott S Grieshaber
University of Idaho
Biological Sciences
Moscow, ID 83844

Re: mSystems00053-23R1 (**Computational Modeling of the Chlamydial Developmental Cycle Reveals a Potential Role for Asymmetric Division**)

Dear Dr. Scott S Grieshaber:

Your manuscript has been accepted, and I am forwarding it to the ASM Journals Department for publication. For your reference, ASM Journals' address is given below. Before it can be scheduled for publication, your manuscript will be checked by the mSystems production staff to make sure that all elements meet the technical requirements for publication. They will contact you if anything needs to be revised before copyediting and production can begin. Otherwise, you will be notified when your proofs are ready to be viewed.

If you would like to submit a potential Featured Image, please email a file and a short legend to msystems@asmusa.org. Please note that we can only consider images that (i) the authors created or own and (ii) have not been previously published. By submitting, you agree that the image can be used under the same terms as the published article. File requirements: square dimensions (4" x 4"), 300 dpi resolution, RGB colorspace, TIF file format.

We recognize that the video files can become quite large, and so to avoid quality loss ASM suggests sending the video file via <https://www.wetransfer.com/>. When you have a final version of the video and the still ready to share, please send it to mSystems staff at msystems@asmusa.org.

Sincerely,

Neha Garg
Editor, mSystems

Journals Department
E-mail: mSystems@asmusa.org